# The global response to the pandemic: An empirical cluster analysis of policies targeting COVID-19

**Karl Gauffin** [1]*, **Olof Östergren** [1,2], **Agneta Cederström** [1]

**1** Department of Public Health Sciences, Stockholm University, Stockholm, Sweden, **2** Aging Research Center (ARC), Karolinska Institutet, Stockholm, Sweden

* karl.gauffin@su.se

## Abstract

It is well known that countries differed in their response to the COVID-19 pandemic in terms of the timing and intensity of specific measures such as lockdowns, face masks and vaccine rollout. However, previous studies have not investigated systematic differences in the overall pandemic strategies. We use daily data from the Oxford COVID-19 Government Response Tracker (OxCGRT), between January 2020 and December 2022 focusing on 16 key pandemic policies, including containment, economic, and health system measures, and apply a three-dimensional k-means clustering algorithm to identify distinct overarching strategies based on the type, intensity, and timing of the response adopted by different countries. We identify four distinct strategies; 1) the traditional infectious disease control approach, adopted by a wide range of high- and middle-income countries, which emphasises strict containment policies and movement restrictions, 2) the public health-oriented approach, adopted by developed welfare states with ageing populations and high health care expenditures, which is more flexible over time and focuses more on economic and health policies, such as income support and testing strategies, with less emphasis on stringent containment, 3) high stringency with gradual relaxation, and 4) reactive policies at a minimal level, both adopted by less democratic low- and middle income countries with substantial inequalities and with younger and less vulnerable populations. The findings contribute to understanding how different countries adapted to the pandemic and how these responses may relate to broader socio-political contexts, including welfare state arrangements and economic resilience.

## Introduction

COVID-19 was declared to be a public health emergency of international concern by the World Health Organization (WHO) from January 2020 to May 2023. COVID-19 was by no means the most infectious or deadliest pandemic in history [1,2], but never

**Data availability statement:** The Oxford Covid-19 Government Response Tracker (OxCGRT) is an open dataset including information on which pandemic response measures were enacted by governments, as well as the timing of these responses. URL to Github repository: https://github.com/OxCGRT/covid-policy-dataset?tab=readme-ov-file

**Funding:** Funding for the work on this study was provided by the Swedish Research Council for Health, Working Life and Welfare (project no. 2022 00262).

**Competing interests:** No authors have competing interests.

before has the world population experienced such a massive political response to an infectious disease. Governments and policy makers were given the unenviable task of managing an emerging deadly health threat while simultaneously protecting the population, state and economy from adverse indirect effects of the pandemic management. Countries used a range of measures, with variation in timing, duration and intensity, to achieve this goal. In contrast to earlier pandemics, there was a wealth of publicly available data on cases, hospital care and deaths, as well as indicators of policy measures and compliance. This allowed the public to follow the unfolding pandemic in near-real time, probably contributing to the politicisation and the heated debates around measures like face masks [3], lockdowns [4,5] and medical interventions, including vaccines [6].

The responses of different countries have been compared and scrutinised with regard to their political, social, economic and epidemiological impact. Since the early days of the pandemic, much attention has been paid to specific policy approaches, both in public discourse and in the scientific community. Some studies have focused on individual measures, such as face mask mandates or restrictions on public gatherings [7,8], while others have combined the analysis of a few selected interventions or evaluated specific regions or countries [9,10]. However, individual policies are not adopted or rejected in a vacuum, but are likely to be part of a broader strategy. Moreover, the path chosen by different policy makers is likely to depend to a large extent on the context of the country, such as the demographic profile, the underlying health status of the population, the organisation of the health system, trust, the resources available to individuals and policy makers, and the specific institutional and legal instruments that can be activated. When comparing policy responses, it is then important to look beyond specific policies and focus instead on the overarching principles guiding the pandemic response. However, while it is clear that countries differed in terms of *specific policies*, it is less clear whether countries adopted fundamentally different *strategies*.

Much of the research on the global political response to the pandemic is based on the work of an Oxford based research group, which was an early developer of the 'Oxford COVID-19 Government Response Tracker (OxCGRT)' including a systematic set of longitudinal measures of government responses in 181 countries [11]. Although this resource provides comparable data on policy responses in several dimensions, updated daily, it can still be difficult to identify and classify strategies. The difficulty lies in summarising the implementation of multi-dimensional responses, with each country characterised by its dynamic behaviour over time. A number of studies have instead focused on pandemic responses in a handful of case studies [12–15]. Others have compared the average levels of stringency in a large number of countries, but without focusing on the differences in the policy measures taken [16]. Here, we take an inductive approach and instead apply an unsupervised learning algorithm to daily information from 177 countries on the government response index (GRI), which comprises 16 measures, including 8 containment and closure policies, 2 economic policies, and 6 health system policies (S1 Table). This approach allows us to take full advantage of the global, longitudinal and multidimensional nature of the OxCGRT data.

This article examines whether there were distinct policy response strategies, defined as clusters of responses that share characteristics in three respects: (i) the set of specific interventions, (ii) the stringency of the interventions, and (iii) the timing of their implementation. In a second step, we examine the strategies in relation to country-level indicators of economic development, population health, inequalities, democracy, and the number of COVID-19 cases and tests per capita, which indicate the country's ability to control the pandemic.

When interpreting our findings, we use the framework outlined by Diderichsen and colleagues [17] to elucidate the pathways from the social context to health outcomes and for introducing policy interventions at the individual level. The framework highlights how adverse health consequences may disproportionately affect some individuals as a result of differential exposure, differential vulnerability and differential consequences. Applying the Diderichsen framework to a large multi-country analysis may generate new insights into the different ways policy interventions can mitigate health inequities.

## Materials and methods

Data on the government response index and other country level data such as gross domestic product (GDP) per capita, economic inequality, and democracy index are available from the website *Our World in Data* (ourworldindata.org) by open access under the Creative Commons license. In particular, we downloaded daily data of the Oxford COVID-19 Government Response Tracker (OxCGRT) from the 1st of January 2020 to the 31st December 2022, representing 1,096 time points [11]. However, to better align the timing of the responses with epidemiological development, we choose the first day of a reported case as the first time point and followed each country for 1000 days to ensure trajectories of the same length. In total, data from 177 countries were included. Five countries were excluded due to missing data, and an additional four because the first reported case appeared more than a year after the start of the timing. However, these are countries with relatively small populations and excluding them did not affect the results.

## Statistical analysis

In order to identify policy strategies we applied a three-dimensional (3D) k-means clustering algorithm for joint longitudinal data (several time-varying variables) [18]. 3D k-means clustering is an unsupervised learning algorithm that seeks to partition the multi-variate trajectories into $k$ clusters, where the within-cluster variance is minimised. This method is data-driven, and allows us to capture patterns in how different countries responses change over time, identifying distinct temporal trajectories within the dataset. It is an adaptation from the k-means clustering algorithm by considering the multiple variables of interest and their evolution along a shared time dimension, which in this case is the responses to the pandemic evolving day-to-day. We selected a three-dimensional k-means clustering algorithm because it effectively captures multi-policy trajectories over time, clustering countries based on the type, intensity, and timing of interventions. This approach is well-suited for time-series data where multiple interrelated policy measures evolve dynamically. Alternative methods, such as principal component analysis combined with clustering, were considered but were not ideal for capturing both the multivariate structure and temporal evolution of policies. The 3D k-means method ensures that the cluster solution that emerges is data-driven, reflecting natural groupings of pandemic responses rather than imposing predefined number of states or Gaussian distributions in the data. The algorithm proceeds by representing the various countries' responses as a multivariate time series, then by building partitions of the trajectories into the clusters using a 'hill-climbing' algorithm. There is no a priori way of knowing the optimal number of clusters but various diagnostic criteria are computed. Ultimately, 2, 3, 4, 5, and 6 clusters are generated by executing the hill-climbing algorithm 20 times each. The temporal dimension is explicitly handled by measuring the strength of the response each day (alignment) and using a distance metric that reflects not only the proximity of values at any given time point, but also how they evolve over time (time-sensitive similarity). In order to determine the optimal number of clusters, we used five diagnostic criteria including three versions of the Calinski-Habaratz index, as well as the Davies-Bouldin and Ray-Turi indices to see where further increases in the number of clusters results in diminishing returns in terms of variance reduction (S1 Fig).

Thus, algorithms such as 3D k-means clustering aims to identify natural groupings within the dataset. The clusters represent groups of trajectories that exhibit similar patterns of evolution over time across the selected variables. The centroid of each cluster represents the average trajectory, capturing the typical temporal pattern of the observations within that cluster. By analysing these average trajectories, we can draw insights into the underlying temporal dynamics and identify distinct strategies. In the present analyses, the clusters correspond to groups of countries which have followed similar trajectories in how the pandemic was handled across containment, economic, and health system responses.

Our second step was to compare the clusters in terms of pre-pandemic characteristics. From *Our World in Data*, we downloaded data on life expectancy, the proportion of the population over 65 years old, population density, GDP per capita, Gini coefficient for income inequality, health expenditure per capita and democracy index for each country in 2019. We also collected data on the number of COVID-tests per capita throughout the pandemic. In order to present the different measures a comparable scale we standardized the values giving each indicator a mean of 0 and a standard deviation of 1. All data management and statistical analyses was done in R v. 4.4.0, and the kml3d package [19] was used to cluster the countries.

## Results

The cluster analysis revealed four distinct strategies (Fig 1). The first strategy (A) is characterised by a strong emphasis on containment and closure policies. By implementing a range of measures aimed at minimising exposure to disease, this group of countries follows a *traditional infectious disease control* approach. While all but perhaps the last group had a strong initial response to the pandemic with a range of closure and containment measures, the countries in the first group are further characterised by prolonged closure of public transport, stay-at-home requirements, and movement restrictions. In other words, this strategy focused on reducing exposure to the virus by minimising interpersonal contact. For large parts of the world population, the approach taken by this group of countries became a dominant aspect of the pandemic years: a far-reaching limitation of everyday life. Among the countries adopting the second strategy (B), the majority also responded to the pandemic with a number of closure and containment policies. However, compared with the first group, these policies were less pronounced and less sustained. Countries in this group are also characterised by a high level of economic policies, particularly income support for sickness and unemployment. Combined with a higher level of health system policies such as testing, contact tracing and vaccination policies, these countries tend to adopt a *public health-oriented strategy,* focusing on limiting vulnerability to the disease, mitigating the economic consequences of infection, as well as minimising other negative impacts of closure and containment policies. The countries adopting the third strategy (C) follow an approach of *high stringency with gradual relaxation*, similar to the strategy of the first group, but at a less restrictive level. Similar to the first group, these countries focus on closure and containment policies aimed at reducing exposure to the virus, while economic and health system policies are implemented to a lesser extent. Finally, compared to the other countries, the fourth strategy (D) can be characterised as *reactive policies at a minimal level*. There was a notable increase in interventions in early 2022, possibly related to increased infection rates following the emergence of the Omicron variant.

Table 1 provides a summary of the global cluster analysis. A map showing the four different clusters is provided along with a supplementary table listing all the countries divided by cluster (Fig 2 and S2 Table).

### Demographic, economic and political profile of the clusters

The countries following the strategy of *traditional infectious disease control* (A) are heterogeneous in terms of human development, economic prosperity and living standards (Fig 2). These includes the five most populous countries in the world – India, China, the United States, Indonesia and Pakistan – as well as several other countries in the Americas and Asia, making this large cluster the home to some 73 percent of the world's population. There is wide variation across a range of economic and demographic indicators for countries in this cluster, but in general they tend to have fewer people

 

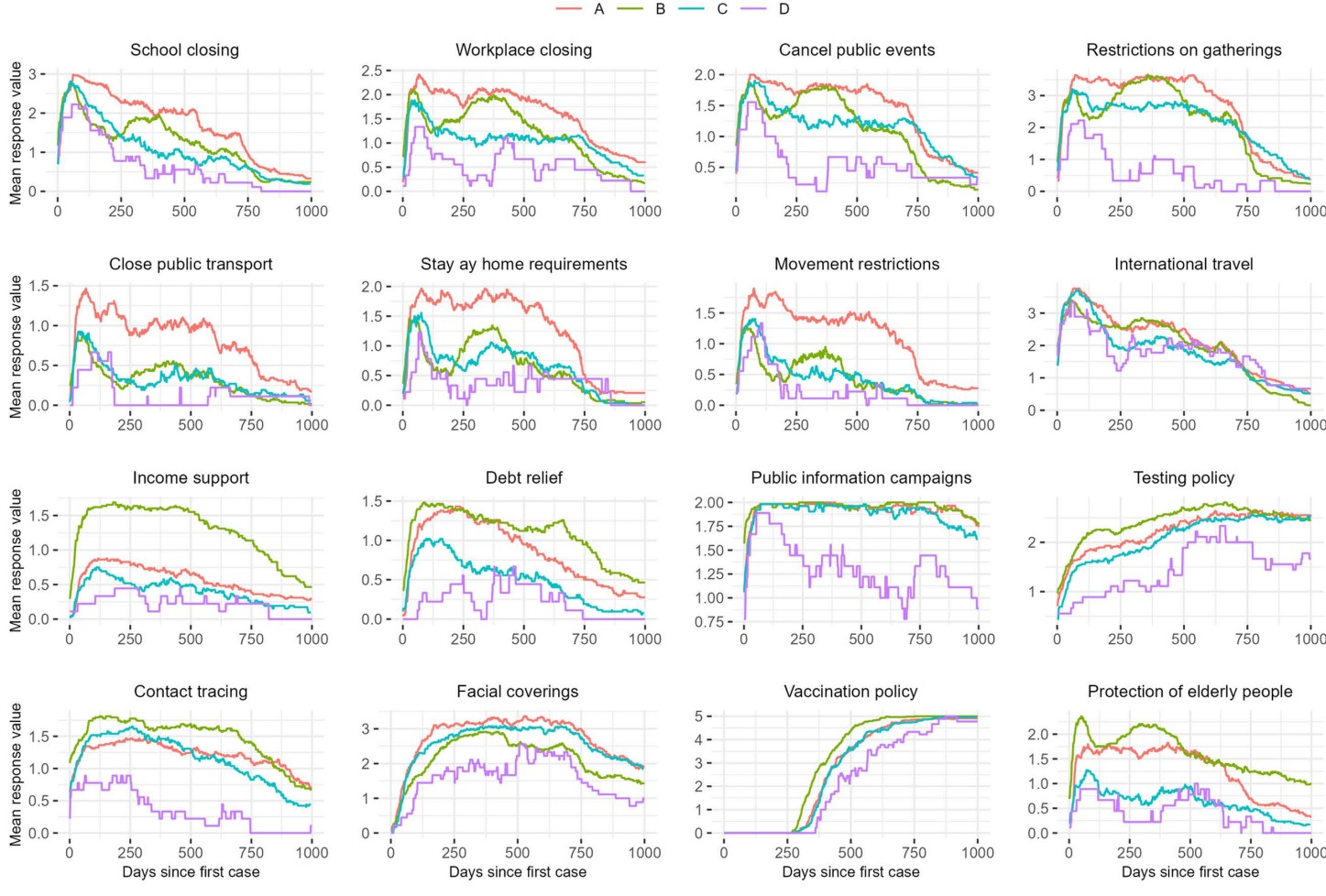

**Fig 1. A global cluster analysis of pandemic responses.**

**Table 1. A summary of the identified strategies.**

|  | A | B | C | D |
|---|---|---|---|---|
| Description | Traditional infectious disease control | Public health-oriented approach | High stringency with gradual relaxation | Reactive policies at a minimal level |
| Type of interventions | Focus on containment and closure policies | Focus on economic and health system policies | Focus on containment and closure policies | Focus on information campaigns and vaccinations |
| Stringency of interventions | High | Medium | Initially high | Low |
| Timing and duration of interventions | Sustained | Flexible | Relaxing | Reactive |

over 65, lower scores on the Democracy Index and fewer tests than countries in Cluster B, although they tend to have more elderly people, a higher Democracy Index and more tests than countries in Clusters C and D. Both the Gini coefficient of income inequality and health expenditure as a share of GDP is more in line with countries in clusters C and D than the high-income countries in the B cluster.

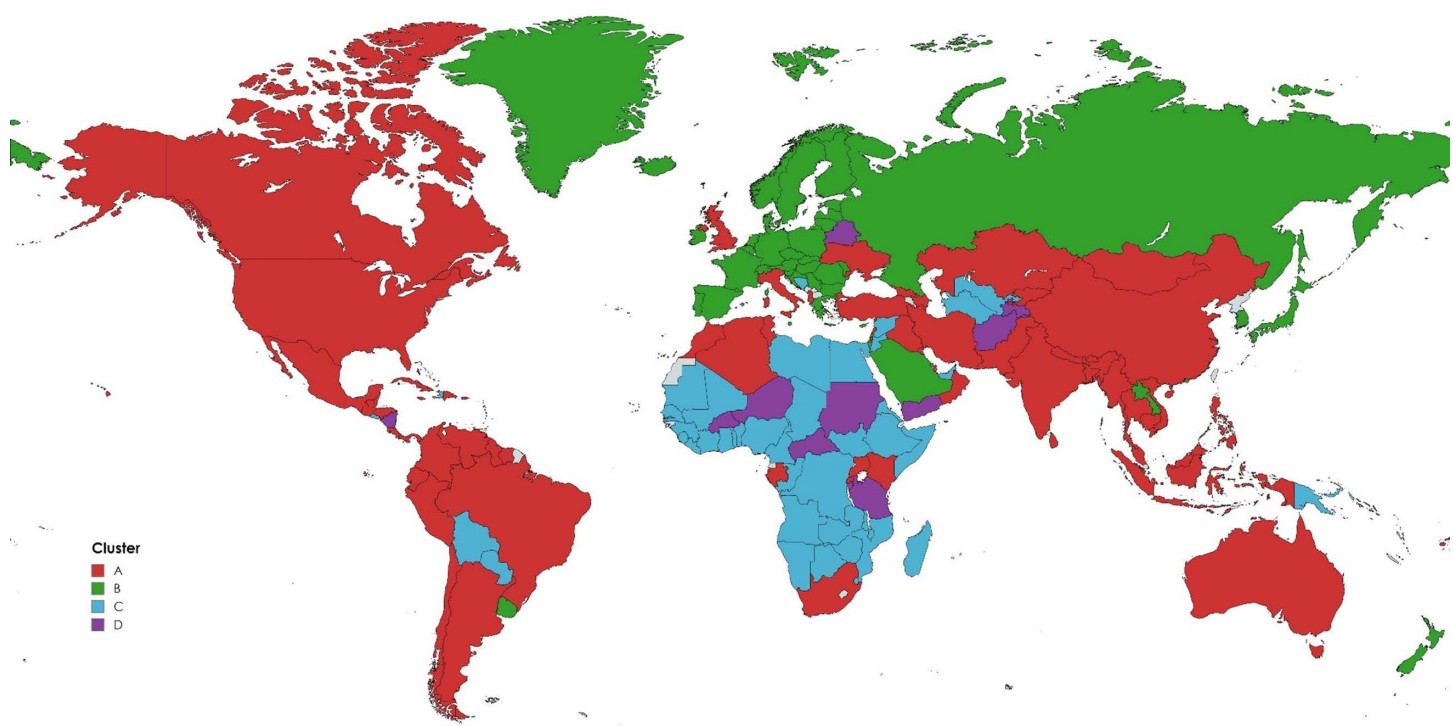

**Fig 2. A world map classifying countries according to pandemic strategy.** Republished from MapChart under a CC BY license, with permission from Minas Giannekis, owner and funder of MapChart, original copyright 2025.

Countries following a *public health-oriented* (B) strategy are home to about 10 percent of the world's population and comprise most European countries, Russia, Saudi Arabia, Japan, South Korea and New Zealand. The countries following this strategy are more affluent compared to the other clusters and have an older population, tend to have lower income inequalities and score high on the democracy index. They also tend to spend comparably more on health care as a percentage of GDP and perform a larger number of tests, indicating that the flexible approach tends to be coupled with a functioning system of disease monitoring.

The countries following a strategy of *high stringency with gradual relaxation* (C) have higher average national income, but are otherwise comparable to the countries in cluster (D), that follow a strategy of *reactive policies at a minimal level*. Cluster (C) includes most of Africa and selected countries in Asia and South America, representing about 13 percent of the world's population while the smaller cluster (D) represents about 4 percent of the world's population and includes countries such as Afghanistan, Belarus, Niger, Sudan, Tanzania and Yemen. The countries in cluster C and D have lower GDP per capita, a younger population, and spend less on health care as percentage of GDP. They also tend to have a high Gini index and score low on the democracy index and performed few COVID-19 tests per capita (Fig 3).

## Discussion

Policy responses to COVID-19 share the explicit goal of protecting public health, but show considerable diversity in the type, intensity and timing of measures. This study has used open access data on 16 different pandemic measures in a global analysis of 177 countries. We used a multivariate trajectory clustering approach using the k-means algorithm to group similar pandemic measures based on their evolution over time across different types of responses. The clustering allows a panoramic differentiation of the pandemic responses in terms of type, intensity and timing of measures, which

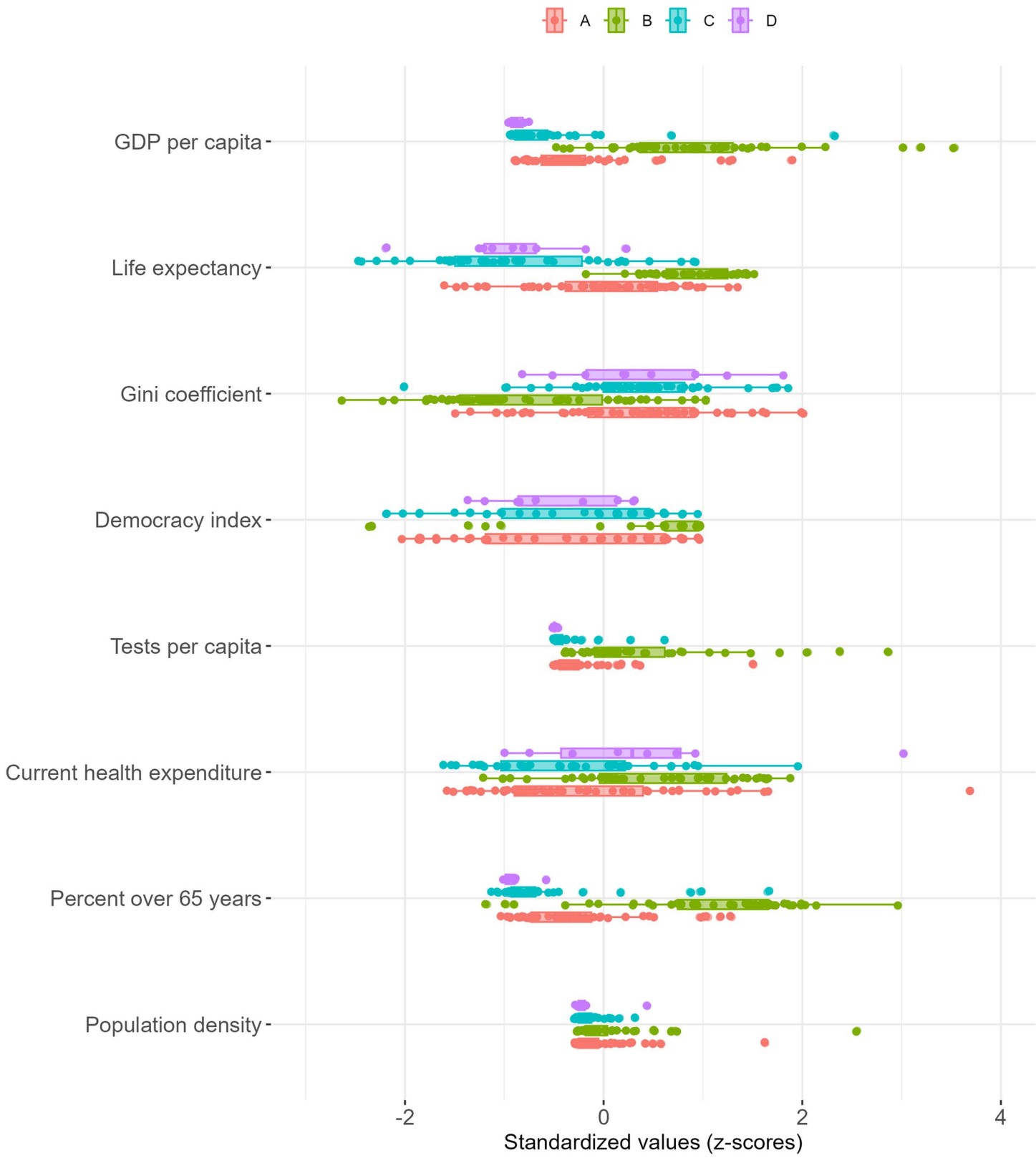

**Fig 3. Distribution of economic, demographic, and health related factors of the four clusters.**

can be used to identify the underlying strategy. The results suggest that there were four distinct strategies adopted in response to the pandemic and that these tended to be chosen by different types of countries. The clustering analysis is based on the overall trajectory of pandemic responses across multiple policy dimensions over nearly three years, rather than on isolated policy decisions at specific points in time. As a result, some country pairings may initially seem unexpected when viewed through the lens of individual policies. However, their broader response patterns – particularly in terms of the intensity and timing of interventions – exhibited sufficient similarity to be grouped within the same cluster.

A major distinction between the pandemic approaches was the focus on containment and closure versus economic and health system policies. Countries with a more traditional infectious diseases approach were more likely to implement containment and closure policies, particularly in terms of movement restrictions, home confinement and public transport closures. Applying the Diderichsen model, this strategy can be characterised as focused on minimising *exposure* to the infectious agent SARS CoV-2, sometimes by preventing people from moving freely, but also by implementing mandates for face-coverings to a greater extent. This strategy had less focus on mitigating *consequences* of the infection in terms of testing and contact tracing, but also in terms of non-health related consequences in the form of income support and debt relief. The countries that followed this strategy were the most diverse in terms of economic standard, democracy level and health-related measures.

The countries that took a more public health-oriented approach also implemented some containment and closure measures, but had a much stronger focus on income support and prolonged debt relief compared with the other countries. In addition, these countries implemented more health system policies, including testing policies, contact tracing, vaccination policies and policies to protect the elderly, to a slightly greater extent than the other countries. While some of these measures reduce the exposure to SARS-CoV-2, their main focus is on *minimising vulnerability* to disease and the *adverse economic consequences* that may result from either illness or the societal impact of the pandemic. Countries following this strategy had a much higher GDP compared to the other groups, invested more of their national income in health care and performed more tests per capita. The success of an adaptive approach to mitigation, where different societal values are likely to be taken into account, depends on accurate information about the spread of the virus in the population, and on a variety of flexible resources that can be activated when and where they are needed.

While this approach could be criticised for vagueness and risking citizen lives during an ongoing deadly pandemic, the countries with a sustained and strict response needed to face the challenges of pandemic fatigue and adverse consequences for the economy and society. Perhaps the most notable example is China and its strict 'zero-COVID-policy' that was relaxed as late as December 2022 following widespread social protest. Sustained and stringent measures to restrict individual mobility and economic activity may then be more difficult to implement successfully in democratic countries. From this perspective, it is notable that many high-income Anglosphere countries, including the United States, the United Kingdom and Australia adopted a strategy focused on containment and closure policies. Contrary to their ideals of individual freedom, these countries have imposed far-reaching restrictions on people's rights to free movement. One could argue that, in theory, a more developed welfare state could be used to guarantee citizens some rights to freedom, even in times of crisis. However, as the cluster of flexible measures also includes a number of less generous welfare states as well as autocratic political systems, this apparent contradiction is not clear cut.

In low- and middle-income countries in clusters C and D, both containment and health system policies were never quite as high, perhaps reflecting a younger and therefore less vulnerable population. These countries tended to be more unequal, scoring low on the democracy index, and had much fewer economic resources to deal with the pandemic. It is also possible that a minimal response was adapted in part because of institutional constraints and lack of resources to adapt a more comprehensive and sustained response.

## Practical implications for pandemic response in the future

Our analysis of differences in the type, intensity, and timing of pandemic responses reveals significant variation in how countries around the world managed COVID-19. From a public health perspective, it is reasonable to highlight the benefits

of the approach taken by countries in Cluster B – combining decisive containment and closure policies with targeted strategies to reduce vulnerability to the virus. These countries prioritised contact tracing, vaccination programs, widespread testing, and the protection of the elderly. This comprehensive strategy may have also contributed to a reduced reliance on prolonged strict lockdowns, striking a balance between public health safety and individual freedoms. However, it is essential to acknowledge that not all countries had the capacity to adopt this approach. Many lacked the well-developed welfare systems, financial resources, and health infrastructure necessary to act proactively in the face of a pandemic. Additionally, political systems and societal norms play a crucial role in shaping how much control governments can exercise over their populations in times of crisis [20].

The COVID-19 pandemic has, hopefully, provided valuable knowledge and experience that will help countries respond more effectively to future outbreaks. During the pandemic, the Independent Panel for Pandemic Preparedness and Response, appointed by the World Health Organization (WHO), criticized the lack of global preparedness and called for an improved surveillance and alert system, as well as greater independence, authority, and funding for the WHO and other global health governance structures [21]. Since then, however, new crises have emerged – including armed conflicts and the U.S. government's decision to cut development aid and withdraw from the WHO – raising concerns about the future of international pandemic cooperation.

It is impossible to determine a single 'best' pandemic response – there is no one-size-fits-all solution. Nevertheless, sustained vigilance and preparedness will be critical in facing future pandemics. This, however, cannot be the responsibility of individual countries alone – it must remain a priority for the global community.

## Methodological considerations

Although the clustering indicates different strategies adopted in response to the pandemic, the categorisation of countries is not precise enough to draw any definitive conclusions about how the economic, democratic, and health system profile of a given country predicts which type of pandemic response that will be implemented. The specific conditions of each country are likely to have had a profound impact on the nature of the response, and efforts to understand specific countries should take these into account. This study focuses on the dimensions of type, intensity, and timing, as these are the aspects systematically captured by the OxCGRT dataset. However, future research should explore additional dimensions such as policy scope, enforcement, communication, compliance, equity, and resource intensity, which could provide deeper insights into the effectiveness and societal impact of pandemic responses. The development of more comprehensive cross-national datasets incorporating these factors would be a valuable contribution to the field. In addition, while changes in political leadership during the pandemic may have influenced policy decisions in some countries, our clustering approach captures the overall trajectory of pandemic responses over time, irrespective of whether shifts were driven by leadership transitions, epidemiological developments, or broader political and socioeconomic factors. Although leadership changes, such as in the United States, were associated with notable shifts in pandemic policies, our findings suggest that structural factors, including welfare state arrangements and economic resilience, played a more central role in shaping national response strategies.

We did not consider whether the different response patterns were associated with systematic differences in epidemiological indicators of disease transmission or on population health. We made this choice for both conceptual and technical reasons. It is difficult to disentangle the response from the course of the pandemic because policy responses depend to some extent on how governments respond to the course of the pandemic. In addition, the consequences of a particular response depend on the specific conditions of the context in which it is applied [22,23]. A true test of the appropriateness of each response pattern cannot then be assessed by comparing countries, as the 'best' response is likely to differ between countries with different conditions, but rather by adapting a counterfactual approach that compares different approaches in the same context. Finally, comparable data on case numbers, hospitalisations and deaths are not available at the global level. Excess mortality has been identified as a good comparative measure because it requires relatively few

data points and does not rely on test or cause of death statistics [24]. However, even small random errors in the registration of vital events can lead to sizeable errors in late-life mortality rates [25,26], which means that measures based on vital statistics, such as excess mortality, are not comparable on a global scale. Taken together, these factors suggest that any differences in health outcomes between response patterns would be difficult to interpret. Studies that aim to evaluate the effectiveness of policy responses at the global level must carefully consider these conceptual and technical issues. While we believe such an exercise would be both intuitive and informative, it is beyond the scope of this article.

## Concluding remarks

Beyond specific individual mitigation measures, we show that countries adopted fundamentally different strategies in the face of the COVID-19 pandemic. The type of strategy differed between countries with different levels of population health, economic development, political profile and ability to effectively monitor the pandemic.

Countries with a more developed welfare state and a large vulnerable population were able to implement stringent and flexible responses, while those with less developed health systems will rely more on sustained containment policies focused on limiting exposure through interpersonal contacts. Countries with fewer resources and smaller vulnerable populations implemented less stringent and sustained responses, possibly due to reduced need and economic capacity.

## Supporting information

**S1 Table. Government response index.**
(DOCX)

**S2 Table. Countries included in the cluster analysis.**
(DOCX)

**S1 Fig. Determination of optimal number of clusters.**
(TIF)

## Acknowledgments

The world map shown in Fig 2 was published from under a CC BY license, with permission from MapChart, original copyright 2025.

## Author contributions

**Conceptualization:** Karl Gauffin, Olof Östergren, Agneta Cederström.

**Formal analysis:** Karl Gauffin, Olof Östergren, Agneta Cederström.

**Methodology:** Agneta Cederström.

**Project administration:** Karl Gauffin.

**Visualization:** Agneta Cederström.

**Writing – original draft:** Karl Gauffin.

**Writing – review & editing:** Karl Gauffin, Olof Östergren, Agneta Cederström.

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
