## [Decision Letter · Decision Letter 0]

21 Jan 2025

PONE-D-24-48926The global response to the pandemic: an empirical cluster analysis of policies targeting COVID-19PLOS ONE

Dear Dr. Gauffin,

Thank you for submitting your manuscript to PLOS ONE. After careful consideration, we feel that it has merit but does not fully meet PLOS ONE’s publication criteria as it currently stands. Therefore, we invite you to submit a revised version of the manuscript that addresses the points raised during the review process.

We look forward to receiving your revised manuscript.

Kind regards,

Anton Pak

Academic Editor

PLOS ONE

Journal Requirements:

 “Funding for the work on this study was provided by the Swedish Research Council for Health, Working Life and Welfare (project no. 2022 00262)”

“Funding for the work on this study was provided by the Swedish Research Council for Health, Working Life and Welfare (project no. 2022 00262)”

“Funding for the work on this study was provided by the Swedish Research Council for Health, Working Life and Welfare (project no. 2022 00262)”

5. We note that [Figure 2] in your submission contain [map/satellite] images which may be copyrighted. All PLOS content is published under the Creative Commons Attribution License (CC BY 4.0), which means that the manuscript, images, and Supporting Information files will be freely available online, and any third party is permitted to access, download, copy, distribute, and use these materials in any way, even commercially, with proper attribution. For these reasons, we cannot publish previously copyrighted maps or satellite images created using proprietary data, such as Google software (Google Maps, Street View, and Earth). For more information, see our copyright guidelines: http://journals.plos.org/plosone/s/licenses-and-copyright.

6. Please include a copy of Table 2 which you refer to in your text on page 6.

7. We note you have included a table to which you do not refer in the text of your manuscript. Please ensure that you refer to Table 1 in your text; if accepted, production will need this reference to link the reader to the Table.

8. Please include captions for your Supporting Information files at the end of your manuscript, and update any in-text citations to match accordingly. Please see our Supporting Information guidelines for more information: http://journals.plos.org/plosone/s/supporting-information .

Additional Editor Comments:

I found the study to have a potential to provide relevant contribution to the field. However, there are several areas where the manuscript could be improved or clarified to strengthen its impact:

Methodology: It would be beneficial to include a stronger rationale and justification for the use of a three-dimensional k-means clustering algorithm. Specifically, why this method was chosen over other clustering techniques, and how its application aligns with the study objectives, should be addressed more explicitly.

Practical implications: It would be good to clearly highlighting the practical implications of your research. Providing more explicit examples of how the findings could influence policy or practice would help bridge the gap between theoretical insights and real-world application.

Results: Consider exploring and discussing whether the observed differences in strategies are associated with better or worse performance in terms of epidemiological criteria. Identifying and analysing such patterns could significantly enhance the utility and applicability of the findings.

Figure quality: The quality of the figures should be improved for better readability and visual impact. Ensure that all figures are clear, professional, and effectively convey the intended information.

Rationale for dimensions: Provide a more detailed rationale for selecting type, intensity, and timing as your dimensions of analysis. How do these dimensions align with or advance the existing literature? Discuss whether other potential domains, such as scope (geography, demography), enforcement, communication, compliance, equity, or resource intensity, were considered and justify their exclusion if relevant.

Reviewers' comments:

Reviewer's Responses to Questions

**Comments to the Author**

1. Is the manuscript technically sound, and do the data support the conclusions?

Reviewer #1: Partly

Reviewer #2: Partly

2. Has the statistical analysis been performed appropriately and rigorously? 

Reviewer #1: No

Reviewer #2: No

3. Have the authors made all data underlying the findings in their manuscript fully available?

Reviewer #1: Yes

Reviewer #2: Yes

4. Is the manuscript presented in an intelligible fashion and written in standard English?

Reviewer #1: Yes

Reviewer #2: Yes

5. Review Comments to the Author

Reviewer #1: Understanding the trajectories of responses to COVID-19 is an interesting and important topic for research (it is, in fact, a question that I have spent some time thinking about too) and this paper takes a stab at it. However, I have some concerns about the statistical approach and assumptions that make me question how things were done.

In particular, I think there is a hidden assumption in their approach that limits the usefulness of 3d k-means clustering. My issue can best be seen by the increases one observes in stringency in late 2021 and early 2022 when Omicron was first spreading. Essentially, these policies can be or are dynamic and respond to changes in the incidence of disease. As a result, the clustering algorithm ALSO imposes a uniformity of incidence patterns across clusters. There are alternatives that the authors could take, principally by introducing tools like dynamic time warping that allow for differences in the timing (but not the sequence) of interventions. I would like to see the authors take such an approach to generate clusters that are less reliant on the precise timing of changes in policies, but rather relies on the sequence of policies.

I am also concerned that the choice to present four clusters is arbitrary and the authors have provided no justification for doing so. Typically one would present a skree plot or some similar metrics to justify a particular number of clusters but I could not find any such plot in the paper or supplemental materials. As a result, I am unsure what to make of the results. This issue is exacerbated when I look at the country map and there are things that don't really pass the smell test. How can New Zealand, for example, be in the same class as Sweden? Two countries that exhibited very different policy responses on a number of margins.

I would love to see this analysis broken into two steps, rather than the current one step approach. First, cluster the daily profile of policies (and possibly present the results in a heat map). Then use dynamic time warping and 3d k-means clustering to group the daily policy clusters into the trajectory clusters.

As a final side note--there were changes in political leadership across countries during this time period that may have affected the policy choices, but again there is an assumption that these changes in leadership do not affect the trajectory cluster.

Reviewer #2: Dear Editor

I appreciate the opportunity to review the paper titled: The global response to the pandemic: an empirical cluster analysis of policies targeting COVID-19 for the potential publication in PLOS ONE. I would recommend a major revision for this manuscript. Please see my comment below:

Method session:

- It is unclear to the reviewer how countries were clustered into the four strategy groups. Were the 3D k-means used for clustering policy strategies that simultaneously clustered county groups?

- The authors used a novel approach for grouping countries using 3D k-means clustering based on the policy strategies. As policies throughout COVID change frequently in each country, have the authors considered re-clustering the country groups based on different times of COVID? The clusters might vary at the beginning of COVID compared to Omicron and late COVID time.

- Were there any missing values in the data? If so, how were missing values being handled?

- Was 4 clusters the optimal number of clusters identified by the Calinski-Habaratz index?

- Have the authors considered running a sensitivities analysis on different numbers of clutters and comparing the temporal patterns of the government response index?

Results session:

- Figure 1: It wasn’t clear to the reviewer what the y-axis “index_mean” means. Please clarify.

- The authors stated that “the majority of countries adopting the second strategy B.. ” . Based on the supplementary material, it seems that group A had the highest number of countries.

- Table 1: Could you please provide more information regarding the clustering of the strategies? Of the 16 government response indexes, which are the ones for groups A, B, C, and D? Would you be able to generate a summary statistic of the total number of types of interventions and the mean and SD for the timing and duration of interventions, please?

- Figure 3: The authors might consider a violin plot for trying to show the distribution of each group and facet by demographic, economic and political profile. It is unclear to the reviewer what the x axis “scaled values” mean. Also, figure 3 was not mentioned in the results.

Discussion session:

- The authors may consider referencing: Lyu S, Qian C, McIntyre A, Lee CH. One Pandemic, Two Solutions: Comparing the U.S.-China Response and Health Priorities to COVID-19 from the Perspective of "Two Types of Control". Healthcare (Basel). 2023 Jun 26;11(13):1848. doi: 10.3390/healthcare11131848. PMID: 37444682; PMCID: PMC10341116.

6. PLOS authors have the option to publish the peer review history of their article (what does this mean? ). If published, this will include your full peer review and any attached files.

**Do you want your identity to be public for this peer review?** For information about this choice, including consent withdrawal, please see our Privacy Policy .

Reviewer #1: **Yes: ** Martin Andersen

Reviewer #2: No

---

## [Author Response · Author response to Decision Letter 1]

12 Mar 2025

please see attached document ("Response to reviewers")

---

## [Editor Report · Decision Letter 1]

27 Mar 2025

The global response to the pandemic: an empirical cluster analysis of policies targeting COVID-19

PONE-D-24-48926R1

Dear Dr. Gauffin,

We’re pleased to inform you that your manuscript has been judged scientifically suitable for publication and will be formally accepted for publication once it meets all outstanding technical requirements.

Kind regards,

Anton Pak

Academic Editor

PLOS ONE
---

## [Editor Report · Acceptance letter]

PONE-D-24-48926R1

PLOS ONE

Dear Dr. Gauffin,

I'm pleased to inform you that your manuscript has been deemed suitable for publication in PLOS ONE. Congratulations! Your manuscript is now being handed over to our production team.

Kind regards,

on behalf of

Dr. Anton Pak

Academic Editor

PLOS ONE